# The Effect of Nattokinase-Monascus Supplements on Dyslipidemia: A Four-Month Randomized, Double-Blind, Placebo-Controlled Clinical Trial

**DOI:** 10.3390/nu15194239

**Published:** 2023-09-30

**Authors:** Xiaoming Liu, Xuejiao Zeng, Jinli Mahe, Kai Guo, Panpan He, Qianwen Yang, Zhiwei Zhang, Zhongxia Li, Di Wang, Zheqing Zhang, Lei Wang, Lipeng Jing

**Affiliations:** 1Institute of Epidemiology and Statistics, School of Public Health, Lanzhou University, Lanzhou 730000, China; liuxiaoming313@163.com (X.L.); zengxjlzu@163.com (X.Z.); 17865195791@163.com (J.M.); guokai6117@163.com (K.G.); hepanpan0518@163.com (P.H.); yqw18709186692@163.com (Q.Y.); zzw982022@163.com (Z.Z.); 2BYHEALTH Institute of Nutrition & Health, No.3 Kehui 3rd Street, No.99 Kexue Avenue Central, Huangpu District, Guangzhou 510663, China; lizx2@by-health.com (Z.L.); wangd7@by-health.com (D.W.); 3Department of Nutrition and Food Hygiene, Guangdong Provincial Key Laboratory of Tropical Disease Research, School of Public Health, Southern Medical University, Guangzhou 510515, China; zzqaa501@smu.edu.cn; 4Department of Neurology, Lanzhou University Second Hospital, Lanzhou 730030, China

**Keywords:** dyslipidemia, lovastatin, nattokinase, red yeast rice, nattokinase-monascus supplements

## Abstract

Dyslipidemia, a condition implying high cardiovascular risks, has been widely studied on its potential nutrition interventions, including functional foods. This study aims to examine the effect of nattokinase monascus supplements (NMSs) on cardiovascular biomarkers and carotid intima-media thickness (CIMT) in patients with dyslipidemia. A total of 113 eligible subjects were randomly assigned to receive either NMSs or a placebo (55 and 58, respectively). After a 120-day intervention, there were significant mean absolute changes in total cholesterol (TC), low-density cholesterol (LDL-C), non-high-density cholesterol (non-HDL-C), and low-density cholesterol to high-density cholesterol ratio (LDL-C to HDL-C ratio), with values of −0.52 (95% CI: −0.51 to −0.54) mmol/L, −0.43 (95% CI: −0.45 to −0.41) mmol/L, −0.52 (95% CI: −0.52 to −0.52) mmol/L, and −0.29 (95% CI: −0.30 to −0.28) mmol/L, respectively, between the two groups. However, no significant differences were found in triglycerides (TGs), high-density cholesterol (HDL-C), and CIMT. Furthermore, the results for lipids and CIMT remained essentially unchanged after adjusting for various confounding factors using the analysis of covariance model. There were no significant differences in coagulation, liver function, renal function, or other indicators. No intervention-related adverse events, such as mouth ulcers, drooling, and stomach pain, were reported. The study results demonstrate that NMSs can ameliorate lipid levels (TC, LDL-C, non-HDL-C, and the LDL-C to HDL-C ratio) without the occurrence of adverse events. However, it did not significantly affect serum TG, HDL-C, and CIMT.

## 1. Introduction

Hyperlipidemia is a significant contributor to the development of atherosclerosis-associated vascular disorders, including coronary artery disease and stroke [1,2], which are leading causes of long-term disability and mortality worldwide, causing an estimated 4.4 million deaths annually [3]. In China, the prevalence of dyslipidemia is 31.2% [4], while the awareness rate of dyslipidemia is only 5.9% [5]. Over the past decades, the extensive use of effective drugs such as statins, ezetimibe, bile acid sequestrants, and PCSK9 inhibitors has been instrumental in managing this condition [6]. However, statins can have side effects such as muscle pain, new-onset diabetes mellitus, and hemorrhagic stroke [7]. Due to these reasons, nearly 30% of people discontinue statin therapy due to muscle soreness [8]. The use of statins in low-risk subjects remains controversial. In recent years, there has been a research focus in recent years on developing safer alternative treatments for lowering serum cholesterol and economical, effective functional foods to improve hyperlipidemia [4,9].

Natto, a fermented soybean product from China, has been used for many years with rare safety disputes [10]. It contains biologically active substances like nattokinase that can improve blood lipids and vascular and fibrinolytic activity. However, an increase in blood uric acid concentration after soybean consumption has been reported in some individuals [11]. Red yeast rice (RYR) is a natural extract obtained via the fermentation of white rice with the yeast monascus purpureus mold and is widely used to reduce serum cholesterol in patients. Monascus can produce biologically active substances with hypolipidemic effects, such as lovastatin (monacolin K) and gamma-aminobutyric acid [12]. However, the toxin citrinin, found naturally in RYR, has been associated with hepatotoxicity and nephrotoxicity [13]. In addition, intakes of RYR failed to produce monacolins in certain subgroups of the population [9,14]. 

Although several studies on single-strain fermentation have been widely studied [1,15,16,17], research on mixed-strain fermentation and its safety is limited in research [14]. In a 4–16 week intervention involving 804 participants from France, Taiwan, USA, Japan, and Italy between 1999 and 2013, red yeast rice combined with policosanol, artichoke leaf extract, berberine, and vitamin C effectively reduces TG, TC, and LDL-C levels with a daily dose of 2.5–10 mg monacolin K [18]. Furthermore, most existing research only provides evidence on natto or RYR’s cholesterol-lowering effects separately [15,16,17,18,19,20], but there is limited evidence regarding the combined NMSs’ (a fermented product of the natto containing nattokinase and RYR containing lovastatin) effects as well as its safety.

Therefore, we conducted a randomized, double-blind, placebo-controlled trial to examine the effect and safety of NMSs on patients with dyslipidemia.

## 2. Materials and Methods

### 2.1. Study Design and Participants

This double-blind, placebo-controlled, randomized trial was conducted in Jing Yuan County Hospital of Traditional Chinese Medicine, Jingyuan County, Baiyin City, Gansu Province, China, from April 2021 to December 2021 to recruit eligible participants and administer the intervention. Treatment allocation remains masked for patients who subsequently enrolled in the ongoing phase 3 extension study according to age and gender. 

Inclusion criteria are (1) being aged 30–70 years old; voluntary participation; and (2) having dyslipidemia condition, with TC ≥ 5.2 mmol/L and TG ≥ 1.7 mmol/L. Eligibility required at least two measurements within a 3-month period, with one meeting the criteria. Exclusion criteria included (1) refusal to sign the informed consent; (2) undergoing pregnancy or lactation; (3) allergic reaction to the sample; (4) serious diseases of the heart, liver, renal, hematopoietic system, or psychiatric disorders; and (5) use of medications that are known to regulate lipid within 3 months or others that may affect the effectiveness of the trial.

According to previous research,52 cases per group were needed [21]. The power analyses of the primary and secondary outcomes for the intention-to-treat population are shown in Appendix A.

The trial received approval from the Ethics Committee of the School of Public Health of Southern Medical University and is registered with the Chinese Clinical Trials Registry (No. ChiCRCT20190111). In accordance with the requirements of the Human Subject Protection Review Committee, all subjects were informed about the potential rare side effects of liver and muscle toxicity associated with cholesterol-lowering drugs. The study followed the CONSORT guidelines for reporting randomized controlled trials. The completed checklist and flow diagram can be found in Appendix A.

### 2.2. Randomization and Masking

In this study, 113 patients were randomly assigned to either the NMS or placebo group for a 120-day intervention (55 and 58 patients, respectively). Patients, investigators, and the sponsors remained masked to the randomization assignment for the duration of the study.

### 2.3. Procedures

A total of 2300 volunteers were initially assessed at Jing Yuan County Hospital of Traditional Chinese Medicine, and 113 eligible patients were enrolled. Patients received oral natto and monascus capsules (975 mg twice daily, NMS (a fermented product containing nattokinase 2500 fibrinolytic units (FUs) derived from natto and lovastatin 7 mg derived from RYR, NMS group); or matching placebo (975 mg twice daily, main containing starch, placebo group) for 120 days (55 and 58, respectively). NMSs and placebo were identical in taste, smell, and appearance. All staff and researchers were blinded to the group assignments until the end of the experiment. 

Subjects were asked to return for a questionnaire (including food frequency and physical activity), physical examination, electrocardiogram (ECG), and fasting blood drawings at baseline and at two follow-up times. Carotid artery thickness B-ultrasound was completed at baseline and at 120 days. Blood samples were tested in the hospital laboratory. Biochemical analyses were performed using Beckman Coulter AU680 automatic biochemical analyzer or Mindray CL-2000i. Blood routines were tested in the Mindray BC-900. Coagulations were tested in Coatron 3000, Germany. Non-HDL-C = TC − HDL-C [22]. Subjects were instructed to follow a regular diet and engage in restful activities three days prior to each blood sample collection. The laboratory meets all standards of accuracy and precision required by the program. The average coefficient of variation is less than 10.0%.

The safety of our subjects during the study was monitored through laboratory assessments and physical examinations, including regular measurements of blood uric acid, aspartate transaminase (AST), alanine transaminase (ALT), coagulation, and other blood biochemical and routine data. Any adverse effects reported during interviews with professional researchers were also documented. Adverse drug reactions were judged by investigators according to Karch and Lasagna’s 5 criteria of adverse reactions. Adverse events were defined as those occurring after the first dose of the intervention.

### 2.4. Outcomes

The primary outcomes were the change in blood lipids (TC, TG, HDL-C, LDL-C, non-HDL-C, and LDL-C to HDL-C ratio) from baseline to 120 days. Secondary outcomes included the changes in CIMT from baseline to 120 days. The safety outcomes included biochemical indicators (e.g., coagulation, liver function, renal function, blood glucose, and blood routine), physical examination parameters (e.g., systolic blood pressure and diastolic blood pressure), and adverse events (e.g., nausea, vomiting, and diarrhea). 

### 2.5. Statistical Analysis

Baseline characteristics were analyzed using χ^2^ test, Fisher’s exact test, Mann–Whitney U test, or two-sample *t* test. Analyses on the treatment effects were conducted in the intention-to-treat population and per-protocol population (Appendix A) using the Mann–Whitney U test or two-sample *t* test. Pairwise comparisons for each variable between baseline and 45 days and between baseline and 120 days were carried out by the paired *t* test or Wilcoxon’s signed-rank test for each treatment group. For lipids and CIMT, separate analyses of covariance (ANCOVA) models were performed, adjusting for multiple confounding factors. Subgroup analyses were also performed for patients with borderline hyperlipidemia, for people aged 30–60, for patients aged 60 years and older, and by patient sex. Safety analyses included all patients who received at least one dose of study treatment.

All statistical tests were two-tailed, and a *p*-value of less than 0.05 was considered statistically significant. Analyses were conducted using R 4.2.2 and SAS (version 9.4) for the ANCOVA model.

## 3. Results

### 3.1. Subjects and Compliance Rate

Between 10 April 2021 and 2 August 2021, 113 patients were enrolled in the trial. Of these, 55 patients were randomly assigned to the intervention group and 58 patients to the placebo group. At the conclusion of the study, the average compliance rate for all subjects was 83.19%, with no significant difference between the groups. A total of 49 (89.09%) of 55 patients in the NMS group and 45 (77.59%) of 58 in the placebo group completed the study (Figure 1). 

This study, with a sample size of 55 patients in the NMS group and 58 patients in the placebo group, has a power of 93.91% (Appendix A) at a 0.05 two-sided significance level to detect the absolute change from baseline to 120 days, with a TC absolute change of −0.84 (−1.03 to −0.65) mmol/L in the NMS group and of −0.32 (−0.52 to −0.12) mmol/L in the placebo group after a 120-day intervention.

### 3.2. Baseline Characteristics

Baseline characteristics are shown in Table 1. All characteristics were similar between the two study groups except for tea (*p* = 0.026), with a higher percentage of subjects in the placebo group having high consumption.

### 3.3. Primary and Secondary Outcomes

The baseline values of biomarkers and CIMT were comparable between the two groups. In the NMS group, significant improvements in all lipid outcomes were observed at both the 45-day and 120-day marks. Conversely, in the placebo group, only TC, TG, non-HDL-C, and the LDL-C to HDL-C ratio showed significant improvements at both time points (as shown in Table 2, Figure 2, and Appendix A). In the between-group comparison, significant reductions in TC, LDL-C, non-HDL-C, and LDL-C to HDL-C ratio were observed at both the 45-day and 120-day marks with NMS treatment compared to the placebo (Table 2, Figure 2, and Appendix A). For TC, the absolute change mean differences between the two groups were −0.59 (−0.61 to −0.57) mmol/L and −0.52 (−0.51 to −0.54) mmol/L at 45-day and 120-day intervention, respectively. For LDL-C, the absolute change mean differences between the two groups at 45 days and 120 days intervention were −0.50 (−0.54 to −0.47) mmol/L and −0.43 (−0.45 to −0.41) mmol/L, respectively. For non-HDL-C, the absolute change mean differences between the two groups were −0.62 (−0.65 to −0.59) mmol/L and −0.52 (−0.52 to −0.52) mmol/L at 45 days and 120 days intervention, respectively. For the LDL-C to HDL-C ratio, the absolute change mean differences between the two groups were −0.39 (−0.43 to −0.36) % and −0.29 (−0.30 to −0.28) % at 45 days and 120 days intervention, respectively. There were no significant differences from baseline to 120 days in the absolute changes and percentage changes in the TG, HDL-C, and CIMT between the two groups. 

### 3.4. Laboratory Values and Physical Examination Findings of Interest

There were no significant differences in the following parameters between the two groups at baseline, 45 days, and 120 days (*p* > 0.05) (Appendix A): platelet count (PLT), prothrombin time (PT), international normalized ratio (INR), activated partial prothrombin time (APTT), fibrinogen (FIB), total bilirubin, direct bilirubin, indirect bilirubin, ALT, AST, ALT/AST ratio, γ-glutamyl transpeptidase (γ-GT), albumin (ALB), globulin (GLB), ALB/GLB ratio, urea, uric acid, fasting glucose, white blood cell count, absolute neutrophil count (ANC), neutrophil percentage, absolute lymphocyte count, lymphocyte percentage, eosinophil count, immature granulocyte count, immature granulocyte percentage, hemoglobin, mean hemoglobin, mean hemoglobin concentration, red blood cell count, hematocrit, mean corpuscular volume (MCV), red blood cell volume distribution width coefficient of variation (RDW-CV), mean platelet volume (MPV), platelet distribution width (PDW), platelet-large cell ratio (P-LCR), body mass index (BMI), systolic blood pressure (SBP), and diastolic blood pressure (DBP).

### 3.5. Adverse Events

No significant ECG abnormalities were observed in any of the patients. No intervention-related adverse effects, including mouth ulcers, glossitis, dry mouth, drooling, stomach pain, bloating, abdominal sounds, belching, loss of appetite, food loss, hunger, acid reflux, nausea, vomiting, hematemesis, constipation, dry stool, diarrhea, loose stool, blood in stool, pus and blood in stool, stomachache, and other discomforts, were reported in any of the 113 subjects randomly assigned.

### 3.6. Dietary and Physical Activity

There were no significant differences in the consumption of milk, dairy and soy nut products, plant foods, cereals and potatoes, vegetables and fruit, and metabolic levels between the two groups at baseline, 45 days, and 120 days (Appendix A).

### 3.7. Multi-Factor Analysis

The multivariate-adjusted analysis was essentially consistent with the univariate analysis, except that the TG levels showed no significant difference between the two groups at 120 days after the intervention (*p* > 0.05), as shown in (Table 3).

### 3.8. Subgroup Analysis

In patients with borderline mixed hyperlipidemia, for TC values ranging from 5.2 to 6.2 and TG values from 1.7 to 2.3, the findings were in line with prior analyses (Appendix A). Subgroup analyses by age for those aged 30–60 years, those aged 60 years and older (Appendix A), and subgroup analyses by sex were generally consistent with the previous results (Appendix A).

## 4. Discussion

This randomized controlled trial, conducted over a period of four months with adults aged 30–70 years old suffering from dyslipidemia, indicated that the consumption of NMSs led to a favorable improvement in lipid levels, suggesting a potential mechanism by which NMSs may reduce cardiovascular risk. However, no significant effect of NMSs on the CIMT was observed after the 120-day intervention. 

### 4.1. Significant Effect on Lipids

In humans, high levels of TG in the bloodstream have been directly linked to atherosclerosis, a risk factor for heart disease and stroke. It is well established that TC, TG, LDL-C, non-HDL-C, and HDL-C are important factors against cardiovascular events [23]. Previously, the administration of RYR preparations at a daily dosage ranging from 2.5 to 10 mg of monacolin K has been associated with a significant anti-hypocholesterolemia impact, demonstrating a reduction in TC up to 20% [24]. Similarly, multiple clinical trials investigating the effects of red yeast rice [25,26,27,28] consistently demonstrated positive lipid improvements in adults aged 18 to 75 years. However, clinical trials using a combination of red yeast rice and natto as an intervention are still limited.

Our study found that the NMSs significantly (*p* < 0.05) reduced TC by 13.75%, TG by 16.10%, LDL-C by 10.99%, non-HDL-C by 20.18%, and LDL-C to HDL-C ratio by 16.31%, as well as increased the HDL-C levels by 7.03%, but there were no significant differences in HDL-C (*p* > 0.05) after a 120-day administration period. Compared to a 6-month intervention study [29], our study had a smaller lipid-lowering effect, except TG lowering magnitude was higher. This is probably due to their overall worse baseline lipid levels (TC of 5.20 to 7.80 mmol/L and TG of 2.27 to 5.70 mmol/L, LDL-C of 3.38 to 5.20 mmol/L, or HDL-C < 1.00 mmol/L for male and <1.25 mmol/L for female) and smaller sample size (NMS, 19; placebo, 10) compared to ours. After controlling for baseline levels, sex, age, baseline biochemical measurement levels, and other various confounding factors, significant differences were observed in terms of TC, TG, LDL-C, non-HDL-C, and LDL-C to HDL-C ratio levels when comparing the NMS group with the placebo group. These results suggest that a 120-day intake of combined NMSs may have a positive effect on patients with dyslipidemia.

Low HDL-C levels are associated with an elevated risk of cardiovascular events and mortality in CAD patients [22]. Conversely, increases in HDL-C levels can serve as predictors of atherosclerosis regression. Our study aligns with previous research [25,26,29], but it is essential to note that we recruited participants with HDL-C levels within the normal range. Future investigations should prioritize studying individuals with initially abnormal HDL-C levels.

The precise mechanism by which NMSs favorably impact lipid levels remains unclear. Based on these observations, we hypothesize that the beneficial effects of NMSs in patients with dyslipidemia may be due to the combined effect of natto and red yeast rice increasing the activity of monacolin K, which inhibits HMG-CoA reductase, the rate-limiting enzyme for cholesterol synthesis, and thus effectively lowers cholesterol levels [19]. 

### 4.2. Lack of Significant Effect on CIMT

CIMT is a reliable surrogate marker of atherosclerosis and is predictive of clinical cardiovascular events [30]. Clinical trials using 2000 FU of nattokinase have observed a significant effect on variables of CIMT [31]. However, our study did not observe a significant effect of NMSs on variables of CIMT, even after adjusting for sex, BMI, and other confounding factors. Although the progression of subclinical atherosclerosis could potentially be delayed or even reversed by the administration of lipid-lowering drugs or lifestyle modification, a detectable anatomical change in carotid arteries may require a longer intervention period to emerge [32]. Future studies with a longer duration and among adults with dyslipidemia are warranted.

### 4.3. Safety of NMSs in Humans

Our study reported no intervention-related adverse results throughout the investigation period, including liver function, fasting blood sugar, coagulation function, and renal function, such as blood uric acid. Although there was a between-group difference in the change in PT from baseline to 45 days, this difference was no longer statistically significant at the 120-day mark, and the PT levels remained in the reference range in both groups during the study period. Further study may be needed to study the potential mechanism. Questionnaire evaluations also revealed no notable adverse events caused by the NMS. Furthermore, dietary intake, metabolic level, and physical examination were evenly distributed between the two groups. These results support the safety of NMS supplements.

In summary, we found that supplementation of NMSs significantly lowered lipid levels in patients with dyslipidemia. The results suggest that NMSs could be a safe and effective nutraceutical for reducing cardiovascular disease risk factors in patients with dyslipidemia, compared to the placebo group.

### 4.4. Strengths

We examined long-term indicators of CIMT, as well as coagulation function and liver and kidney function, to comprehensively assess the impact of NMS supplementation on various aspects of cardiovascular and overall health. In addition to the above, dietary habits, metabolic profiles, and participants’ medical histories were collected and incorporated into the multifactorial analysis. Remarkably, the results remained stable and robust, further affirming the potential benefits of NMS supplementation in the context of cardiovascular health. Given the low rate of effective hyperlipidemia control, our study provides promising evidence that the use of dietary supplements presents a safe and cost-effective approach to addressing this health concern.

### 4.5. Limitations

Our study has several limitations to consider. This study was conducted at a single center with a relatively short intervention period of 120 days. To gain a more comprehensive understanding of lipid-lowering efficacy and safety, it would be beneficial to conduct longer-term interventions with multiple research centers in future studies. The sample size in our study was relatively small, which may limit the generalizability of the findings. In addition, we lacked in vivo marker concentrations of nutritional supplements in the blood, and we did not collect lipid testing data at 120 days post-intervention, which could evaluate the sustainability of the observed effects. We did not include a control group for the lovastatin drug group, which prevented us from comparing NMSs’ efficacy with conventional drug interventions. These limitations should be taken into consideration when interpreting the results and designing future research in this area.

## 5. Conclusions

In conclusion, our study provides evidence that NMSs have a favorable effect on the reduction of TC, LDL-C, non-HDL-C, and LDL-C to HDL-C ratio compared with a starch placebo in dyslipidemia, although no significant effect was observed in TG, HDL-C, and CIMT. No drug-related adverse events were observed. This fills a gap in the research of nutritional health products. These preliminary results need to be confirmed in future research. Additional studies are essential to elucidate potential underlying mechanisms. The results may represent a novel nutraceutical intervention in dyslipidemia management for patients who do not meet the clinical requirements of statin use.

## Figures and Tables

**Figure 1 nutrients-15-04239-f001:**
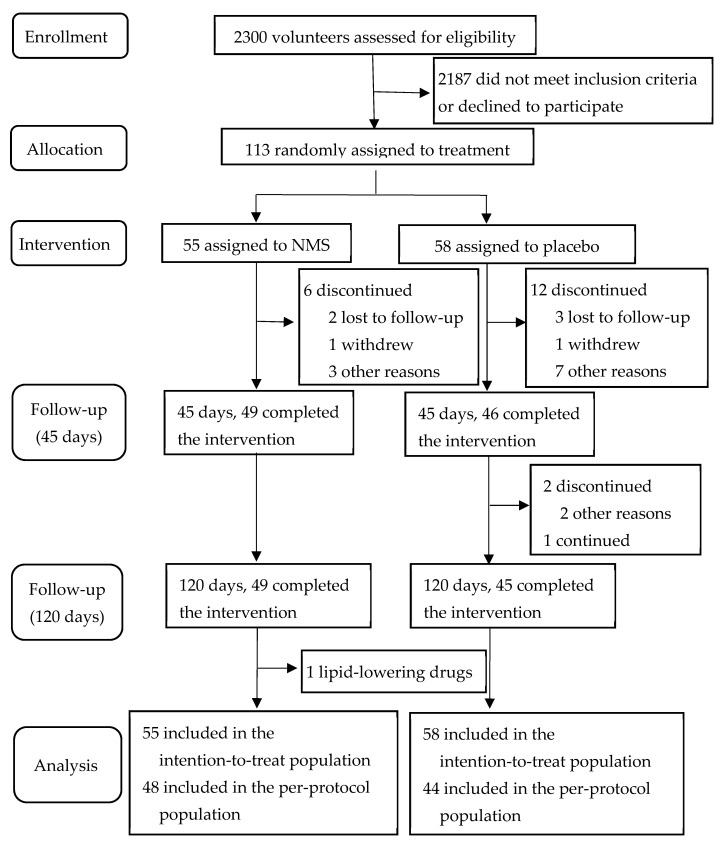
Screening, allocation, and study follow-up.

**Figure 2 nutrients-15-04239-f002:**
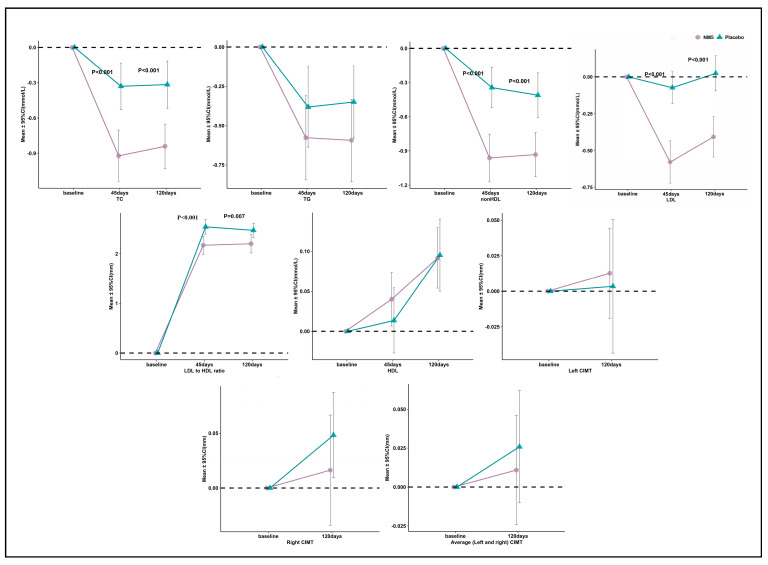
Change in lipids at baseline, 45 days, and 120 days.

**Table 1 nutrients-15-04239-t001:** Baseline characteristics.

Characteristics	NMS (*n* = 55)	Placebo (*n* = 58)	*p*
Age (year)	58.47 ± 8.81	59.79 ± 9.10	0.228
Male/Female (n)	14 (25.5)/41 (74.5)	13 (22.4)/45 (77.6)	0.705
Smoking (n)	7 (12.7)	10 (17.2)	0.502
Alcohol (n)	5 (9.1)	5 (8.6)	1.000
Lipid-lowering drugs (n)	0	0	
Fish oil (n)	3 (5.5)	3 (5.2)	1.000
Other supplements (n)	27 (49.1)	23 (39.7)	0.313
Traditional Chinese medicine (n)	7 (12.7)	6 (10.3)	0.692
Tea (n)	18 (32.7)	31 (53.4)	0.026
Medical history (n)	44 (80.0)	44 (75.9)	0.596
Hypertension (n)	23 (41.8)	28 (48.3)	0.491
Diabetes (n)	8 (14.5)	8 (13.8)	0.909
Heart disease (n)	7(12.7)	4.0 (6.9)	0.296
Chronic hepatitis (n)	0	0	
Chronic renal insufficiency (n)	0 (0.0)	1 (1.7)	1.000
Kidney ureter or bladder stones (n)	3 (5.5)	1 (1.7)	0.355
BMI (kg/m^2^)	26.17 (25.43 to 26.91)	26.61 (25.84 to 27.38)	0.412
SBP (mmHg)	132.09 (127.47 to 136.71)	132.71 (127.98 to 137.43)	0.852
DBP (mmHg)	85.07 (82.15 to 87.99)	83.47 (80.71 to 86.22)	0.424
Waist-to-hip ratio (%)	93.54 (92.09 to 94.99)	94.18 (92.53 to 95.83)	0.561
Milk, dairy, and soy nut products (times/day)	0.80 (0.58 to 1.03)	0.76 (0.58 to 0.94)	0.724
Animal food (times/day)	1.22 (0.95 to 1.48)	1.34 (1.06 to 1.63)	0.609
Plant foods (times/day)	7.37 (6.58 to 8.16)	7.38 (6.73 to 8.03)	0.886
Cereals and potatoes (times/day)	3.70 (3.35 to 4.05)	3.49 (3.22 to 3.76)	0.499
Vegetables and fruits (times/day)	3.67 (3.07 to 4.26)	3.89 (3.38 to 4.40)	0.629
Metabolic levels (kcal/day)	2132.78 (1880.76 to 2384.79)	1954.52 (1785.49 to 2123.56)	0.526

*p* values were calculated by χ^2^ tests, Fisher exact tests, Mann–Whitney U tests, or two-sample *t* tests between the two groups of different variables. Data are mean ± SD, mean (95% CI), or patient numbers with percentages in the table. Smoking, more than 5 packs of cigarettes cumulatively in a year; alcohol, at least one alcoholic drink per week for 6 consecutive months; lipid-lowering drugs, taken at least once in the last three months; fish oil, taken an average of ≥1 capsule per month; other supplements, including calcium tablets and vitamin supplements, taken more than 30 times cumulatively in the past year; tea, at least 2 times per week, changing tea leaves twice; medical history and other illnesses, hospital diagnoses in lifetime; SBP, systolic blood pressure; DBP, diastolic blood pressure; and BMI, body mass index.

**Table 2 nutrients-15-04239-t002:** Primary and secondary outcomes in the intention-to-treat population.

End Outcomes	NMS (*n* = 55)	Placebo (*n* = 58)	NMS vs. Placebo	
	Mean (95% CI)		Mean Difference (95% CI)	*p* Value
**Primary outcomes**				
**TC**				
**Baseline, mmol/L**	5.99 (5.77 to 6.20)	5.86 (5.62 to 6.10)	0.13 (0.15 to 0.10)	0.469
**45 days, mmol/L**	5.06 (4.87 to 5.26) ***	5.53 (5.35 to 5.70) **	−0.46 (−0.49 to −0.44)	<0.001
**Absolute** **change from baseline to 45 days, mmol/L**	−0.93 (−1.14 to −0.70)	−0.33 (−0.53 to −0.13)	−0.59 (−0.61 to −0.57)	<0.001
**Percentage change from baseline to 45 days, %**	−14.75 (−18.04 to −11.46)	−4.40 (−7.66 to −1.14)	−10.35 (−10.38 to −10.33)	<0.001
**120 days, mmol/L**	5.14 (4.92 to 5.37) ***	5.54 (5.37 to 5.71) **	−0.40 (−0.45 to −0.34)	0.006
**Absolute change from baseline to 120 days, mmol/L**	−0.85 (−1.03 to −0.65)	−0.32 (−0.52 to −0.12)	−0.52 (−0.51 to −0.54)	<0.001
**Percentage change from baseline to 120 days, %**	−13.75 (−16.81 to −10.69)	−4.16 (−7.27 to −1.05)	−9.58 (−9.54 to −9.63)	<0.001
**TG**				
**Baseline, mmol/L**	2.66 (2.36 to 2.96)	2.74 (2.47 to 3.01)	−0.08 (−0.12 to −0.04)	0.430
**45 days, mmol/L**	2.08 (1.78 to 2.39) ***	2.36 (2.12 to 2.60) **	−0.28 (−0.34 to −0.21)	0.020
**Absolute change from baseline to 45 days, mmol/L**	−0.58 (−0.84 to −0.31)	−0.38 (−0.64 to −0.13)	−0.20 (−0.21 to −0.18)	0.169
**Percentage change from baseline to 45 days, %**	−17.56 (−28.64 to −6.48)	−8.88 (−17.59 to −0.18)	−8.67 (−11.05 to −6.30)	0.045
**120 days, mmol/L**	2.07 (1.83 to 2.30) ***	2.39 (2.15 to 2.63) ***	−0.32 (−0.32 to −0.33)	0.033
**Absolute change from baseline to 120 days, mmol/L**	−0.59 (−0.85 to −0.33)	−0.35 (−0.58 to −0.12)	−0.24 (−0.28 to −0.21)	0.120
**Percentage change from baseline to 120 days, %**	−16.10 (−27.19 to −5.01)	−7.48 (−16.38 to 1.42)	−8.61 (−10.80 to −6.43)	0.058
**LDL-C**				
**Baseline, mmol/L**	3.66 (3.49 to 3.84)	3.56 (3.40 to 3.72)	0.10 (0.09 to 0.11)	0.568
**45 days, mmol/L**	3.09 (2.92 to 3.26) ***	3.49 (3.36 to 3.61)	−0.40 (−0.45 to −0.36)	<0.001
**Absolute change from baseline to 45 days, mmol/L**	−0.57 (−0.72 to −0.43)	−0.07 (−0.18 to 0.03)	−0.50 (−0.54 to −0.47)	<0.001
**Percentage change from baseline to 45 days, %**	−15.20 (−18.89 to −11.50)	−0.76 (−3.89 to 2.37)	−14.44 (−15.01 to −13.87)	<0.001
**120 days, mmol/L**	3.26 (3.06 to 3.46) ***	3.59 (3.44 to 3.73)	−0.33 (−0.38 to −0.28)	0.009
**Absolute change from baseline to 120 days, mmol/L**	−0.40 (−0.54 to −0.27)	0.03 (−0.09 to 0.14)	−0.43 (−0.45 to −0.41)	<0.001
**Percentage change from baseline to 120 days, %**	−10.99 (−14.72 to −7.25)	1.83 (−1.57 to 5.23)	−12.82 (−13.15 to −12.48)	<0.001
**non-HDL-C**				
**Baseline, mmol/L**	4.54 (4.32 to 4.75)	4.46 (4.24 to 4.68)	0.08 (0.08 to 0.07)	0.767
**45 days, mmol/L**	3.58 (3.37 to 3.79) ***	4.12 (3.96 to 4.27) ***	−0.54 (−0.60 to −0.48)	<0.001
**Absolute change from baseline to 45 days, mmol/L**	−0.96 (−1.17 to −0.75)	−0.34 (−0.52 to −0.17)	−0.62 (−0.65 to −0.59)	<0.001
**Percentage change from baseline to 45 days, %**	−20.49 (−24.74 to −16.23)	−6.06 (−9.81 to −2.31)	−14.43 (−14.93 to −13.92)	<0.001
**120 days, mmol/L**	3.61 (3.38 to 3.83) ***	4.05 (3.90 to 4.20) ***	−0.44 (−0.52 to −0.37)	0.002
**Absolute change from baseline to 120 days, mmol/L**	−0.93 (−1.13 to −0.74)	−0.41 (−0.61 to −0.22)	−0.52 (−0.52 to −0.52)	<0.001
**Percentage change from baseline to 120 days, %**	−20.18 (−24.26 to −16.09)	−7.45 (−11.28 to −3.63)	−12.72 (−12.98 to −12.47)	<0.001
**LDL-C to HDL-C ratio**				
**Baseline**	2.63 (2.45 to 2.82)	2.61 (2.46 to 2.77)	0.02 (−0.01 to 0.05)	0.888
**45 days**	2.17 (1.99 to 2.35) ***	2.54 (2.39 to 2.69) *	−0.37 (−0.40 to −0.34)	0.001
**Absolute change from baseline to 45 days**	−0.46 (−0.55 to −0.37)	−0.07 (−0.13 to −0.01)	−0.39 (−0.43 to −0.36)	<0.001
**Percentage change from baseline to 45 days, %**	−17.74 (−21.18 to −14.30)	−2.03 (−4.39 to 0.33)	−15.71 (−16.79 to −14.62)	<0.001
**120 days**	2.20 (2.02 to 2.38) ***	2.47 (2.33 to 2.61) **	−0.27 (−0.31 to −0.23)	0.007
**Absolute change from baseline to 120 days**	−0.43 (−0.53 to −0.33)	−0.14 (−0.23 to −0.05)	−0.29 (−0.30 to −0.28)	<0.001
**Percentage change from baseline to 120 days, %**	−16.31 (−20.09 to −12.53)	−4.34 (−7.64 to −1.03)	−11.97 (−12.45 to −11.50)	<0.001
**HDL-C**				
**Baseline, mmol/L**	1.45 (1.37 to 1.53)	1.40 (1.33 to 1.46)	0.05 (0.03 to 0.07)	0.533
**45 days, mmol/L**	1.49 (1.41 to 1.57) *	1.41 (1.35 to 1.47)	0.08 (0.06 to 0.09)	0.233
**Absolute change from baseline to 45 days, mmol/L**	0.04 (0.01 to 0.07)	0.01 (−0.03 to 0.05)	0.03 (0.03 to 0.02)	0.142
**Percentage change from baseline to 45 days, %**	3.28 (0.87 to 5.68)	1.63 (−1.37 to 4.63)	1.64 (2.24 to 1.05)	0.118
**120 days, mmol/L**	1.54 (1.46 to 1.62) ***	1.49 (1.42 to 1.57) ***	0.05 (0.04 to 0.05)	0.428
**Absolute change from baseline to 120 days, mmol/L**	0.09 (0.05 to 0.13)	0.10 (0.05 to 0.14)	0.00 (0.00 to −0.01)	0.713
**Percentage change from baseline to 120 days, %**	7.03 (4.30 to 9.76)	7.29 (4.04 to 10.54)	−0.26 (0.26 to −0.79)	0.758
**Second outcomes**				
**Right CIMT**				
**Baseline, mm**	0.83 (0.78 to 0.89)	0.81 (0.76 to 0.86)	0.02 (0.02 to 0.02)	0.507
**120 days, mm**	0.85 (0.80 to 0.89)	0.86 (0.81 to 0.91) *	−0.01 (−0.01 to −0.01)	0.852
**Absolute change from baseline to 120 days** **, mm**	0.02 (−0.03 to 0.07)	0.05 (0.01 to 0.09)	−0.03 (−0.04 to −0.02)	0.229
**Percentage change from baseline to 120 days, %**	6.48 (−2.17 to 15.13)	8.44 (2.89 to 14.00)	−1.96 (−5.06 to 1.13)	0.234
**Left CIMT**				
**Baseline, mm**	0.88 (0.82 to 0.93)	0.88 (0.82 to 0.94)	0.00 (0.00 to −0.01)	0.951
**120 days, mm**	0.89 (0.83 to 0.94)	0.88 (0.83 to 0.94)	0.00 (0.00 to 0.00)	0.935
**Absolute change from baseline to 120 days** **, mm**	0.01 (−0.02 to 0.04)	0.00 (−0.04 to 0.05)	0.00 (0.02 to −0.01)	0.858
**Percentage change from baseline to 120 days, %**	2.77 (−1.60 to 7.15)	3.06 (−2.54 to 8.66)	−0.28 (0.94 to −1.51)	0.843
**Average (Left and right) CIMT**				
**Baseline, mm**	0.86 (0.81 to 0.91)	0.85 (0.80 to 0.90)	0.01 (0.01 to 0.01)	0.769
**120 days, mm**	0.87 (0.82 to 0.91)	0.87 (0.82 to 0.92)	−0.01 (0.00 to −0.01)	0.876
**Absolute change from baseline to 120 days** **, mm**	0.01 (−0.02 to 0.05)	0.02 (−0.01 to 0.06)	−0.02 (−0.01 to −0.02)	0.463
**Percentage change from baseline to 120 days, %**	3.00 (−1.77 to 7.78)	4.80 (0.18 to 9.41)	−1.79 (−1.95 to −1.64)	0.464

Mean difference refers to the disparity between two groups. The difference is considered statistically significant when *p* < 0.05, denoted as *; when *p* < 0.01, denoted as **; and when *p* < 0.001, denoted as ***.

**Table 3 nutrients-15-04239-t003:** Adjusted results of ANCOVA model for lipids and CIMT.

Variables	Model	NMS (*n* = 55)	Placebo (*n* = 58)	NMS vs. Placebo	
		Mean (95% CI)		Mean Diff (95% CI)	*p* Value
TC	1	5.40 (5.28 to 5.52)	5.64 (5.53 to 5.76)	−0.24 (−0.08 to −0.41)	0.004
	2	5.95 (4.96 to 6.94)	6.15 (5.16 to 7.15)	−0.20 (−0.03 to −0.37)	0.023
TG	1	2.27 (2.12 to 2.42)	2.50 (2.35 to 2.64)	−0.23 (−0.01 to −0.44)	0.037
	2	2.84 (1.56 to 4.12)	3.03 (1.74 to 4.32)	−0.19 (0.03 to −0.41)	0.095
LDL-C	1	3.34 (3.24 to 3.43)	3.55 (3.45 to 3.64)	−0.21 (−0.08 to −0.34)	0.002
	2	3.10 (2.31 to 3.89)	3.28 (2.49 to 4.07)	−0.18 (−0.05 to −0.32)	0.009
Non-HDL-C	1	3.88 (3.77 to 4.00)	4.18 (4.07 to 4.29)	−0.29 (−0.14 to −0.45)	<0.001
	2	3.91 (2.99 to 4.83)	4.15 (3.23 to 5.08)	−0.24 (−0.09 to −0.40)	0.003
LDL-C to HDL-C ratio	1	2.33 (2.24 to 2.43)	2.54 (2.45 to 2.63)	−0.21 (−0.07 to −0.34)	0.003
	2	1.65 (0.93 to 2.37)	1.83 (1.10 to 2.55)	−0.18 (−0.05 to −0.30)	0.005
HDL-C	1	1.49 (1.45 to 1.53)	1.43 (1.39 to 1.47)	0.06 (0.12 to 0.00)	0.056
	2	2.01 (1.68 to 2.33)	1.96 (1.63 to 2.29)	0.05 (0.11 to −0.01)	0.076
left cimt	1	0.88 (0.84 to 0.92)	0.88 (0.84 to 0.92)	0.00 (0.06 to −0.06)	0.999
	2	0.78 (0.52 to 1.04)	0.77 (0.51 to 1.04)	0.01 (0.06 to −0.05)	0.838
right cimt	1	0.84 (0.81 to 0.88)	0.84 (0.80 to 0.87)	0.00 (0.05 to −0.04)	0.850
	2	0.74 (0.51 to 0.97)	0.73 (0.50 to 0.96)	0.02 (0.06 to −0.03)	0.520
average cimt	1	0.86 (0.83 to 0.90)	0.86 (0.83 to 0.89)	0.00 (0.05 to −0.05)	0.953
	2	0.75 (0.53 to 0.98)	0.75 (0.52 to 0.97)	0.01 (0.05 to −0.04)	0.688

*p* value means the comparison of the two groups; mean diff means mean difference between the two groups; Model 1: no calibration; Model 2: correction: gender, age, BMI, smoking, alcohol, fish oil, lower-lipids drugs, chronic disease, animal’s foods, plants foods, and metabolic levels.

## Data Availability

Not applicable.

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
