# Peer review of "The Effect of Nattokinase-Monascus Supplements on Dyslipidemia: A Four-Month Randomized, Double-Blind, Placebo-Controlled Clinical Trial"

_nutrients, 2023, doi:10.3390/nu15194239_

Round 1

Reviewer 1 Report

The article "The Effect of Nattokinase-monascus Supplements on 2 Dyslipidemia: A Four-Month Randomized, Double-Blind, Placebo-Controlled Clinical Trial", by Xiaoming Liu et al. is an interesting point of view regarding the potential treatment for dylipidemia. The natural compound nattokinase-monascus it seems to be effective on TC, LDL-C, non-HDL-C, and the LDL-C to HDL-C ratio but has no effect on TG, HDL-C and CIMT.

The study is well done, with a clear protocol and results, respecting the ethical recommendations. The results are clear and sustain the main objective of the study.The discussion and conclusions are also clear and well done.

I recommend revising the typographical errors  and results :

line 49: capitalize "red" and also provide information about the evolution of urc acid during the intervention. 

Reviewer 2 Report

This manuscript tested the effect of the NMS supplement in dyslipidemia. The findings are interesting and significant as the supplement reduced total cholesterol and LDL with no effects seen for HDL or TG. As a major risk factor for cardiovascular disease, the reduction in cholesterol is relevant. However, the manuscript needs revision in the following sections:

1)     From the introduction it is suggested that NMS is a fermented product of the natto and RYR, but it was not explained. Then, in line 102-103 it is mentioned that NMS is nattokinase plus lovastatin. Why was the RYR mentioned?

2)     The age-range of 30-70 is broad. Table 1 shows that the average age is 58.47±8.81 and 59.79 ±9.1 in the NMS and placebo groups, respectively. The low SD suggest that many of the participants were in the higher end of the age range. How many participants were in the 30-40, 40-50, 50-60 range and 60-70? The sub analysis for 30-60 and >60 seems odd. What is the average age for the 30-60 group?

3)     Sub analysis for sex should be also shown.

4)     For the number of participants, only the participant completing the intervention should be counted. For the 56 and 58 mentioned in the abstract, 48 and 45 completed the intervention based on Fig. 1

5)     Revisions are needed for the final numbers for the placebo group in Fig. 1. 45 participants completed the intervention with 44 used for analysis.

6)     In line 104, the number of participants that completed the intervention should be used (48, 44) instead of the ones initially enrolled (55, 58). The same for the tables showing the outcomes.

7)     Line 135, if 52 cases per group were needed, then this study did not meet the criteria.

8)     All data in Table 2 should be revised, for example, the absolute change from baseline to 45 days for TC for the NMS group is 0.93 not 0.92, for LDLL-C is 0.57 instead of 0.58.

9)     Legend for Figure 2 is missing.

Needs a few revisions

Round 2

Reviewer 2 Report

All of my concerns were satisfactorily addressed by authors 

Author Response

Dear Reviewer,
   We appreciate your acknowledgment that your concerns have been fully addressed. We are committed to ensuring the quality and accuracy of our article and we are pleased that the revisions we made have met your expectations. Once again, we sincerely appreciate your valuable time and professional insights.